

# The impact of advance organizers in virtual classrooms on the development of integrated science process skills

Abdellah Ibrahim Mohammed Elfeky[1], Ali Hassan Najmi[2] and Marwa Yasien Helmy Elbyaly[3]

[1] Department of Curriculum and Instruction, College of Education, Najran University, Najran, Saudi Arabia
[2] Department of Educational Technology, King Abdulaziz University, Jeddah, Saudi Arabia
[3] Centre for Sharia, Educational and Humanities Research, Najran University, Najran, Saudi Arabia

## ABSTRACT

Unlike virtual classrooms that have received extensive research attention in both academic and practical contexts because of their ability to improve students' outcomes, the use of advance organizers are still in need for more research to prove their efficacy in fulfilling expected learning outcomes in these virtual classrooms. Hence, the present study aims to identify the impact of using such organizers in virtual classrooms on the development of students' integrated science process skills. The present study was applied to 64 students who were studying for their Master's degree in the vocational education techniques in the "Research Project" course. Participants were randomly divided into two equal experimental groups with 32 students in each. An assessment card of five main domains was used to evaluate students' skills in research procedural definition, identification and control of the research variables, questions and/or hypotheses, procedures and experimentation besides research interpretation of the results. Data analysis showed that the use of advance organizers in virtual classes was of great effect on the development of participants' integrated science process skills because skills of students in the first experimental group were better mastered than the skills of their peers in the second experimental group in accordance with the subskills in the five domains.

Corresponding author
Abdellah Ibrahim Mohammed Elfeky,
abdalah.elfeqi@spe.kfs.edu.eg

## INTRODUCTION

In many nations affected by the COVID-19 pandemic, the situation in higher education institutions has changed (*Elbyaly & Elfeky, 2022*). Therefore, students and teachers have been recommended to switch to online learning instead of face-to-face classes as usually occurs at various educational institutions (*Basilaia & Kvavadze, 2020*). One type of this online learning is the virtual classroom that shares some characteristics with ordinary or traditional classrooms (*Basilaia & Kvavadze, 2020*; *Fitton, Finnegan & Proulx, 2020*) such as flexibility, practicality, and accessibility to learning environments that are not limited by time or place (*Elbourhamy, Najmi & Elfeky, 2023*; *Elbyaly & Elfeky, 2023*). Najran University in Saudi Arabia, as well as other educational institutions in other countries,

could transfer to online learning and conduct virtual classrooms by providing its staff with an educational platform known as 'Blackboard', a system that enabled both students and faculty members to restore the whole content and requirements of any academic course. Synchronous and asynchronous discussions are provided through this system in addition to the ability to conduct the electronic tests and utilize many multimedia files (*Elfeky & Elbyaly, 2023*). Moreover, virtual classes are allowed on this Blackboard system as one tool of e-learning.

Virtual classrooms are electronic classrooms that can be protracted in time, space, and content (*Ricolfi, 2020*; *Far, Ferron & Ibarra, 2015*). They offer various learning environments that enable video and audio broadcasts besides discussion boards (*Ruthotto, Kreth & Stevens, 2020*). They are virtual because they can overcome spatial constraints where learners can attend from different locations without time constraints. One of the main characteristics of virtual classrooms is the fact that virtual sessions can be recorded in advance and then delivered to students who can watch them everywhere at their own pace and time (*Ahmed, Alharbi & Elfeky, 2022*; *Capuccini, Larsson & Carone, 2019*). Another characteristic of virtual classrooms is the fact that they are used in the educational process to cancel any geographical barriers where students can enter the class and record the whole session. Moreover, they reduce students' anxiety, increase their motivation, and more importantly improve their communication and collaboration (*Suwais & Alshahrani, 2018*; *Elbyaly & El-Fawakhry, 2016*).

In addition, the term of 'advance organizers' was first used by Ausubel in the 1960s (*Suwais & Alshahrani, 2018*). They represent the introductory material to be presented to learners before the teaching content (*Vander Meij, 2019*). They are used to familiarize students with new concepts and connections between them before beginning integrating new knowledge to their prior information (*Nisyah et al., 2020*; *Men, Bryan-Kinns & Bryce, 2019*). In brief, advance organizers are used when asking students to recall and combine existing knowledge with new information offered in various learning situations (*Han-Chin & Hsueh-Hua, 2017*). An advanced organizer is mainly used to activate the learner's prior knowledge to better understand its similarity to new information under study (*Zheng, Yang & Garcia, 2008*). Furthermore, it is used to stimulate students to practice the learning task by creating meaning for them because its use can clarify the learning objective and provide a context to motivate the learner to learn how to accomplish the task (*Masada, 2017*). The use of advance organizers provides learners with commendable advantages that can positively influence students' acquisition of concepts (*McDonald & Vines, 2019*) and involves a set of procedures starting with providing previous knowledge, establishing clear links between prior and new knowledge, teaching new vocabulary in the same context, and highlighting key vocabulary (*Elfeky & Elbyaly, 2021*).

Science process skills, the present study aims to test its improvement through the use of advanced organizers in virtual classrooms, are certain potentials used to enable students to think as scientists and researchers while conducting their research (*Chan & Morales, 2017*; *Alshammary & Alhalafawy, 2023*; *Alzahrani & Alhalafawy, 2023*). They are of two basic models, *i.e.,* basic skills and skills of integrative science processes. Basic skills model includes prediction, measurement, classification, and observation while

integrative science processes model involves procedural definition, variables' identification and control, research questions and hypotheses, procedures and experimentation, and lastly interpretation of results (*Elbyaly & Elfeky, 2022*). Students' empowerment in these integrated science process skills is very important because they help students easily look for new information, solve problems in different situations, and gain knowledge from practice (*Sermsirikarnjana, Kiddee & Pupat, 2017*). Being proficient in such skills, a student can process and build any scientific information and meanwhile understand the nature of the kind of science he is studying (*Duruk, Akgün & Dogan, 2017*; *Najmi, Alhalafawy & Zaki, 2023*). Apart from the various ways and means to access scientific information, advance organizers help students to think scientifically (*Yumusak, 2016*; *Alanzi & Alhalafawy, 2022*; *Alzahrani, Alshammary & Alhalafawy, 2022*). In addition, at Najran University, many academic courses primarily depend on integrated science process skills. One of these courses is the "Research Project" course that assumes that fulfilling students' needs could be through developing these integrated science process skills.

## REVIEW OF RELATED LITERATURE

Empirical studies in accordance with the effect of using advance organizers in classrooms, as an approach of providing educational materials to learners, have proved that they increase academic achievement, enhance ability to learn, and lead to better retention of the learnt materials (*Popova, Kirschner & Joiner, 2014*; *Elfeky, Alharbi & Ahmed, 2022*; *Almalki & Elfeky, 2022*). For example, *Babaei & Izadpanah (2019)* concluded that advance organizers use was effective in improving students' listening comprehension. *Vander Meij (2019)* argued that advance organizers integrated in educational videos were effective in developing students' programming skills. In addition, *Nevisi & Hosseinpur (2019)* revealed an effect of using advance organizers in developing students' ability to write an English abstract. *Susan & Oche (2018)* revealed an impact of using such organizers in developing students' attitude towards learning biology.

In light of these findings, it can be argued that too little research has been carried out in KSA to assume that advance organizers will improve integrated science process skills within this context. Therefore, the present study aims to identify the effect of using advance organizers in virtual classrooms on developing students' integrated science process skills, *i.e.,* Procedural definition, research variables define and adjustment, research questions and assignments, procedures and experimentation, and results interpretation. That is, it aims to answer the following main question:

RQ: How effective is using advance organizers in virtual classrooms in developing students' integrated science process skills (*i.e.,* procedural definition, research variables define and adjustment, research questions and assignments, procedures and experimentation, and results interpretation) and in developing these skills as a whole?

## METHODOLOGY

### Research methods

In the present study, multi-dimensional research and data collection approaches are adopted to evaluate participant students' research plans. In order to allow comparison of evidence from multiple streams in a pragmatic approach to empirical research across groups of respondents, quantitative approach was used into this study (*Lo & To, 2023*) to answer its research question. A quantitative approach to data analysis is better used with data that can be examined and quantified for correlations with other sets of data (*Lo, 2023*). This approach is considered appropriate for the use of Likert's five-point scale for students' scores on each subskill of the Integrated Scientific Process skills Evaluation Card where scores are used to compare the level of skills of participants in both groups.

### Data collection

An assessment card of five domains was designed and used to evaluate students integrated science process skills, (Appendix S1A), after reviewing related educational literature and previous studies such as *Elfeky & Elbyaly (2017)*, *Hernawati, Amin & Irawati (2018)*, *Juhji & Nuangchalerm (2020)*, *Sharma (2020)* and *Elfeky & Elbyaly (2016)*. To confirm its reliability, it was offered to a jury of experts and specialists who were faculties at Najran University in the fields of educational technology, curriculum and instruction and psychology. Once they accepted the task, they were requested to express their views with regard to the card's sub-skills or phrases appropriateness to measure each domain that stands for one skill of the integrated science processes. Additionally, they were requested to check the phrases' clarity and soundness of their linguistic formulation, and they were asked to add any items or delete inappropriate ones. Thus, the final version of the card consisted of five main domains *i.e.,* procedural definition domain with five subskills or phrases, study variables identification and control domain with five subskills, research questions and hypotheses domain with six subskills, procedures and experimentation domain with six subskills, and interpretation of results domain with four subskills (see Appendix S1A). Likert's five-point scale was used beginning from 1 = Very little to 5 = Very much for each domain's subskill or phrase. Moreover, Cronbach's Alpha was used to check the card's internal consistency. Stability of the scorecard was verified by applying it to a pilot sample of 30 students who were excluded of the study. Stability coefficient was (0.88) for the card as a whole, indicating that it was fit for the study aims and for evaluating the students' final research plans. Three specialists and impartial faculty members were requested to evaluate each final product of every participant. Once evaluation was done, the three scores of each student were accounted for to work out the average score of each participant.

### Study sample

Sixty-four students who were studying for their Master's degree in vocational education techniques participated in the present study. They were all enrolled in a "Research Project" academic course provided by Najran University in Saudi Arabia. They were all male students; their average age was 23 years old with a standard deviation about 2.48. The participant students were randomly selected and divided into two experimental groups,

*i.e.,* first and second experimental groups. The first experimental group learnt the course *via* the use of advance organizers in virtual classrooms while the second experimental group studied the course *via* virtual classrooms without using any advance organizers. Advance organizers were provided to students in the first experimental group *via* virtual classrooms at the beginning of each lecture before starting in-detail lesson explanation and discussion *via* virtual classrooms.

## Data analysis

ANOVA was used to evaluate students' homogeneity before the beginning of the experiment, *i.e.,* pre-evaluation stage. At the end of the course, *t*-test for independent samples were used to compare the means of students' scores in both groups.

## Ethical considerations

The Najran University Deanship of Scientific Research review board approved conducting the present study in its decision No.: 444-45-22144-DS. Then all participants were informed about the main objective of the present study during the orientation meeting at the beginning of the course. Furthermore, participants who agreed to take part in the study were requested to submit their written consent electronically *via* the Blackboard system. Moreover, it is worth mentioning that methods used in this investigation followed the guidelines set forth in the Helsinki Declaration.

## Pre-assessment of participants' science integrated process skills

At the beginning of the first week of the course (Sunday), each participant was informed *via* an advertisement on the university's Blackboard system to submit a complete research plan for the topic chosen at the end of the weak (Saturday). An independent evaluator who was a faculty member at the college of education evaluated the whole research plan using the developed evaluation card. All scores were collected and processed using ANOVA. Results are presented in Table 1.

Results in Table 1 show that the differences between the mean scores of students in both experimental groups regarding their integrated science process skills in the pre-application card were not statistically significant ($\alpha leq 0.05$). In other words, participant students' skills in accordance with the research plan's five main domains were homogeneous.

## Experimental processing teaching material

Before delivering the course content to students, it was organized into ten lectures to discuss the main research plan domains, *i.e.,* procedural definition, variables identification and control, questions and hypotheses, procedures and experimentation, and interpretation of results. To fully fulfill the main objectives of the present study, many instructional design models were reviewed to come up with a set of indicative steps to follow while designing and producing these lectures. In addition, learners' characteristics were accounted for, too. After that, synchronous virtual classes were used to deliver each lecture to students in both experimental groups using the Collaborate Ultra Experience LTI application developed during the emergence of Covid-19 pandemic and integrated within Blackboard e-learning management system by the Deanship of Information Technologies at Najran University.

**Table 1 ANOVA for the differences between the mean scores of participants in both groups in the evaluation card pre-application.**

| Skills of integrated science process | Source of variance | Sum of squares | DF | Mean of square | F. ratio | Sig. |
|---|---|---|---|---|---|---|
| Procedural definition | Between Groups | 2.976 | 1 | 2.976 | 1.697 | 0.194 |
| | Within Groups | 364.838 | 62 | 1.754 | | |
| | Total | 367.814 | 63 | | | |
| Variables identification and control | Between Groups | 2.743 | 1 | 2.743 | 1.747 | 0.188 |
| | Within Groups | 326.514 | 62 | 1.570 | | |
| | Total | 329.257 | 63 | | | |
| Questions and hypotheses | Between Groups | 0.576 | 1 | 0.579 | 0.279 | 0.598 |
| | Within Groups | 429.619 | 62 | 2.065 | | |
| | Total | 430.195 | 63 | | | |
| Procedures and experimentation | Between Groups | 2.743 | 1 | 2.743 | 1.827 | 0.178 |
| | Within Groups | 312.286 | 62 | 1.501 | | |
| | Total | 315.029 | 63 | | | |
| Interpretation of results | Between Groups | 1.376 | 1 | 1.376 | 1.438 | 0.323 |
| | Within Groups | 199.124 | 62 | 0.957 | | |
| | Total | 200.500 | 63 | | | |
| **Total** | Between Groups | 1.719 | 1 | 1.719 | 0.193 | 0.661 |
| | Within Groups | 1,851.276 | 62 | 8.900 | | |
| | Total | 1,852.995 | 63 | | | |

In fact, such an application can be seen as a unique addition to university instructors who use Blackboard system because it enabled them to hold virtual meetings, live lectures, and video conferencing. In addition, it provided them with a chat room, live audio and video broadcasting, whiteboard, application sharing, synchronous web browsing, and feedback.

Thus, students in the first experimental group studied the course content through the use of virtual classrooms using advance organizers *via* ten lectures, while students in the second experimental group learnt the content *via* the use of virtual classrooms only without using any advance organizers *via* ten lectures also. Specifically, expository advance organizers were utilized to give students some foundational information to form an idea about each newly introduced topic. Moreover, a more official concept map was used to present some crucial components of every new topic. Thus, new concepts were presented in a clearer picture, their origins, definition, and main characteristics. Figure 1 presents a comparison between both experimental groups in accordance with the use of advance organizers.

## RESULTS

Once the experiment ended by the end of the academic semester, dimensional arithmetic mean scores of participant students in both experimental groups regarding their integrated science process skills, as measured by the developed evaluation card, were extracted, and then modified gain ratio was calculated. *T*-tests for independent samples were used to compare the modified gain ratios of students in both groups.

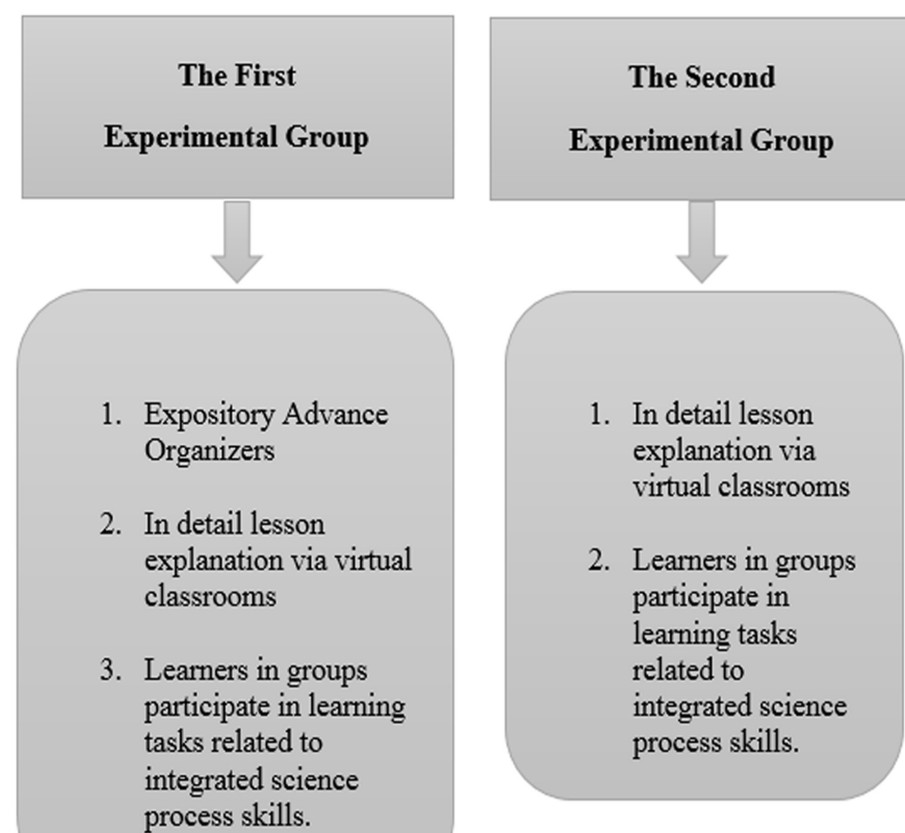

**Figure 1** A comparison between study's groups regarding the use of advance organizers.

Table 2 reveals that the t. value for the difference between the modified gain ratios of participants in both groups regarding the skills of the procedural definition domain was (4.40). It shows that the mean score of students in the first experimental group was ($M = 13.42$), while it was ($M = 11.02$) for peers in the second experimental group. That is, there was a statistically significant difference between participants in both groups regarding their skills in the procedural definition in favor of first experimental group students. This difference ($p = 0.013 < 0.05$) indicates that learners in the first experimental group who studied the course *via* virtual classes with the use of advance organizers outperformed their peers in the second experimental group who studied the course through virtual classes without the use of advance organizers.

To check the soundness of this result, Eta Square ($\eta2$) was used to make sure of the effect size of the virtual classes where advance organizers are used in the procedural definition skills of students in the first experimental group. Findings showed that $\eta2$ calculated significance was (0.361). To put it in different words, there was a significant effect of using advance organizers in virtual classrooms on the development of students' skills in the procedural definition.

**Table 2** *T*-test for the differences between the modified gain ratios of participants' integrated science process in both experimental groups.

| Skills of integrated science process | Group | M | SD | Mean difference | T. Ratio | Sig. |
|---|---|---|---|---|---|---|
| Procedural definition | The first group | 13.42 | 4.271 | 2.40 | 4.40 | 0.013 |
| | The second group | 11.02 | 2.944 | | | |
| Identification and control of variables | The first group | 14.21 | 3.548 | 2.58 | 5.948 | 0.021 |
| | The second group | 11.63 | 2.679 | | | |
| Questions and hypotheses | The first group | 18.02 | 3.322 | 2.86 | 6.90 | 0.034 |
| | The second group | 15.16 | 2.638 | | | |
| Procedures and experimentation | The first group | 17.06 | 3.502 | 2.36 | 5.409 | 0.046 |
| | The second group | 14.70 | 2.784 | | | |
| Interpretation of results | The first group | 11.49 | 2.791 | 1.40 | 3.457 | 0.043 |
| | The second group | 10.09 | 3.052 | | | |
| Total | The first group | 74.19 | 8.557 | 11.6 | 10.608 | 0.036 |
| | The second group | 62.59 | 7.233 | | | |

With regard to participants' skills in variables identification and control domain, Table 2 indicates that the t. value for the difference between the modified gain ratios of participants in both groups regarding the skills of variables identification and control domain was (5.948). It shows that the mean score of students in the first experimental group was ($M = 4.21$), while it was ($M = 11.63$) for peers in the second experimental group. That is, there was a statistically significant difference between participants in both groups regarding their skills in variable identification and control in favor of first experimental group students. This difference ($p = 0.021 < 0.05$) indicates that learners in the first experimental group who studied the course *via* virtual classes with the use of advance organizers outperformed their peers in the second experimental group who studied the course through virtual classes without the use of advance organizers.

To check the validity of this result, Eta Square ($\eta2$) was used to make sure of the effect size of the virtual classes where advance organizers are used in variables identification and control skills of students in the first experimental group. Findings showed that $\eta2$ calculated significance was (0.385). To say it differently, there was a significant effect of using advance organizers in virtual classrooms on the development of students' skills in variables identification and control.

Results regarding participants' skills in writing the research questions and hypotheses domain Table 2 shows that the *t* value for the difference between the modified gain ratios of participants in both groups regarding the skills of stating the research questions and hypotheses domain was (6.90). It shows that the mean score of students in the first experimental group was ($M = 18.02$), whereas it was ($M = 15.16$) for peers in the second experimental group. That is, there was a statistically significant difference between participants in both groups regarding their skills in stating the research questions and hypotheses in favor of first experimental group students. This difference ($p = 0.034 < 0.05$) indicates that learners in the first experimental group who studied the course *via* virtual classes with the use of advance organizers outperformed their peers in the second

experimental group who studied the course through virtual classes without the use of advance organizers.

To ascertain this result, Eta Square ($\eta2$) was used to make sure of the effect size of the virtual classes where advance organizers are used in writing research questions and hypotheses skills of students in the first experimental group. Findings showed that $\eta2$ calculated significance was (0.379). That is, there was a significant effect of using advance organizers in virtual classrooms on the development of students' skills in stating research questions and hypotheses.

With regard to the fourth domain, Table 2 indicates that the t. value for the difference between the modified gain ratios of participants in both groups regarding the skills of research procedures and experimentation was (5.40). It shows that the mean score of students in the first experimental group was ($M = 17.06$), whereas it was ($M = 14.70$) for their mates in the second experimental group. In other words, a statistically significant difference was noticed between participants in both groups regarding their skills in research procedures and experimentation in favor of first experimental group students. This difference ($p = 0.046 < 0.05$) indicates that learners in the first experimental group who studied the course *via* virtual classes with the use of advance organizers outperformed their peers in the second experimental group who studied the course through virtual classes without the use of advance organizers.

To be sure of this result, Eta Square ($\eta2$) was used to make sure of the effect size of the virtual classes where advance organizers are used in research procedures and experimentation skills of students in the first experimental group. Findings showed that $\eta2$ calculated significance was (0.352), which means that there was a significant effect of using advance organizers in virtual classrooms on the development of students' skills in research procedures and experimentation.

With regard to findings related to the last domain, Table 2 shows that t. value for the difference between the modified gain ratios of participants in both groups regarding the skills the interpretation of the research's results was (3.457). It shows that the mean score of students in the first experimental group was ($M = 11.49$), but it was ($M = 10.09$) for their colleagues in the second experimental group. That is, a statistically important difference was found between participants in both groups regarding their skills in research results interpretation in favor of the first experimental group students. This difference ($p = 0.043 < 0.05$) indicates that learners in the first experimental group outperformed their peers in the second experimental group. To test the correctness of this result, Eta Square ($\eta2$) was used to make sure of the effect size of the virtual classes where advance organizers are used in the interpretation of research results skills of students in the first experimental group. Findings showed that $\eta2$ calculated significance was (0.344), showing that a significant effect of using advance organizers in virtual classrooms on the development of students' skills in research results interpretation was found.

Moreover, finding in Table 2 show that t. value for the difference between the skills of participants in both groups related to the evaluation card as a whole was (10.608). The mean score of students' achievement in the first group was ($M = 74.19$), in comparison with their peers' achievement in the second group that was ($M = 62.59$). To say it differently, there

was a statistically significant difference between the mastery levels of students' integrative science process skills in both groups as a whole in favor of the first group. The difference that was ($p = 0.036 < 0.05$) shows that skills of students who studied the course content through the use of advance organizers in the virtual classes were developed more than the skills of their mates in the second experimental group who studied the course content *via* virtual classes but without the use of advance organizers.

To be certain whether this result is correct or not, ($\eta$2) was used to determine the effect size of advance organizers in virtual classes in the learners' integrated science process skills as a whole. Findings prove that the calculated significance of ($\eta$2) for the difference between participants' modified gain ratios in both groups was (0.395). Therefore, it can be argued that utilization of advance organizers in virtual classes have an impact on the development of students' integrated science process skills as a whole.

## DISCUSSION

The main aim of the present study was to investigate the effect of using advance organizers in virtual classrooms on the enhancement of the integrative science process skills of students enrolled in "Research Project" course at Najran University. Findings regarding the development of learners' integrative science process skills in the study main domains namely, procedural definition, research variables define and adjustment, research questions and assignments, procedures and experimentation, and results interpretation. Analysis of data collected *via* the use of an evaluation card shows that there is a significant effect of these advance organizers use in the development of participants' skills in all tested skills and subskills. Mean scores of student's research proposals using the evaluation card indicate the effect of using advance organizers on promoting students' skills in such crucial areas for research project writing. These findings corroborate the results concluded by earlier studies regarding the effect of using advance organizers in the achievement of different learning outcomes. Among these studies was *Birabil (2020)*, which concluded that advance organizers-based teaching strategy could improve learners' academic achievement and retention. Results support the results by *Hadi (2016)* that proved the effectiveness of advance organizers in improving students' gaining and understanding new knowledge. Results also support the results by *Babaei & Izadpanah (2019)* that proved the effectiveness of advance organizers in improving students' listening comprehension. Results also corroborate the results of *Day (2018)* and *Lewis, Chen & Relan (2018)* that proved the effectiveness of advance organizers in students' knowledge construction. They are congruent with the results of *Vander Meij (2019)*, which confirmed the effectiveness of advance organizers included in educational videos in developing students' programming skills. moreover, results are in agreement with the findings of *Nevisi & Hosseinpur (2019)* with regard to the effectiveness of advance organizers in developing learners' ability to write an English abstract and *Susan & Oche (2018)* regarding the impact of advance organizers on the development of students' attitude towards biology.

Results related to the integrated science process skills as a whole were also positive and reflected the effectiveness of using advance organizers in virtual classrooms. *Aktamiş &*

*Ergin (2007)* confirmed that integrated science process skills are more significant than basic skills in learners' epistemic development because they play an important role in developing problem-solving abilities. In short, the skills of students who studied the course content through the use of advance organizers in the virtual classes were developed more than the skills of their mates who studied the course content *via* virtual classes but without the use of advance organizers.

## CONCLUSION AND RECOMMENDATIONS

In the present study, advance organizers, through virtual classrooms attached to Blackboard system provided by Najran University were used to teach students the content of an academic course titled "Research Project". Results indicated that advance organizers were of great effect on enhancing the integrative science process skills mainly the research skills of enrolled students. In light of these findings, it can be claimed that advance organizers utilization in virtual classrooms could be advantageous for students and can develop the processes of instruction and research as well. Therefore, institutions of higher education would benefit from quality teacher training in this area, and future research ought to establish how sufficient this training is in KSA. Researchers, for their part, are recommended to investigate the effect of using advance organizers in other educational contexts in developing different educational outcomes.

### Funding

The authors were funded by the Deanship of Scientific Research at Najran University, under the General Research Funding program grant code NU/DRP/SEHRC/12/13. The funders had no role in study design, data collection and analysis, decision to publish, or preparation of the manuscript.

### Grant Disclosures

The following grant information was disclosed by the authors:
Deanship of Scientific Research at Najran University:  NU/DRP/SEHRC/12/13.

### Competing Interests

The authors declare there are no competing interests.

### Author Contributions

- Abdellah Ibrahim Mohammed Elfeky conceived and designed the experiments, performed the experiments, performed the computation work, authored or reviewed drafts of the article, and approved the final draft.
- Ali Hassan Najmi conceived and designed the experiments, analyzed the data, prepared figures and/or tables, and approved the final draft.
- Marwa Yasien Helmy Elbyaly performed the experiments, analyzed the data, prepared figures and/or tables, authored or reviewed drafts of the article, and approved the final draft.

## Ethics

The following information was supplied relating to ethical approvals (i.e., approving body and any reference numbers):

The Najran University Deanship of Scientific Research review board gave their Approving (No.:444-45-22144-DS).

## Data Availability

The raw data is available in the Supplementary File.

## Supplemental Information

Supplemental information for this article can be found online at http://dx.doi.org/10.7717/peerj-cs.1989#supplemental-information.

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
