# Peer review of "The impact of advance organizers in virtual classrooms on the development of integrated science process skills"

_PeerJ Computer Science, doi:10.7717/peerj-cs.1989_

## Round 0.1 · original submission · Major Revisions

Please follow review comments.

**Language Note:** The review process has identified that the English language must be improved. PeerJ can provide language editing services - please contact us at [email protected] for pricing (be sure to provide your manuscript number and title). Alternatively, you should make your own arrangements to improve the language quality and provide details in your response letter. – PeerJ Staff

Reviewer 1 ·

Basic reporting

The authors should clearly state the objectives of the study or the research questions.
The advanced organises in the context of the research need to be properly explained. This should be done in the methodology section. What advanced organisers were used in the study?
The authors have mixed up the introduction of the study and the section on the review of the literature. This needs to be separated and the introduction section should clearly establish the context of the study.

Experimental design

The research tool used in the study is not very clear. Either the scales and sub-scales should be given as a table in the paper or attached as an appendix for clarity.

Validity of the findings

The findings are ok but very limited as the objective was only to see the difference between the two experimental groups. Further data analysis could also have been done to ensure the richness of the results. Cohen's 'd' should have been calculated to see the effect size, especially after using the 't-test'. How has the impact of advanced organisers in the experimental group been demonstrated?

Additional comments

None

·

Basic reporting

I believe this research meets a standard high enough to be published. However, there are areas that I’ve identified where I believe significant improvements are required.

The paper presents research into advance organisers in virtual classrooms and their impact upon the development of integrated science process skills.

Spelling is generally good but there are some odd word choices throughout and issues with the use of articles as well. I’ve made some minor corrections but the manuscript could do with another thorough proofread so that it reads in a fluid manner.

Academic English is used throughout, though to varying standards. There are some issues with tense in the abstract and the conclusion’s wording is quite confusing. This is an area that requires some attention and potentially third-party proofreading and correction.

The study cites relevant references when claims are made.

The work is generally well-structured, though there appear to be some headings that don’t refer to the text. Perhaps these refer to tables that have not been included or attached. Additionally, more subheadings may help to better structure the work.

Experimental design

There are a lot of pleasing features to the paper. For one, the design seems fundamentally to be a good one with regards to testing the intervention. The explanations of concepts and terms in the introduction is also very clear and assists the reader in understanding the project. However, there are areas where serious improvements are required.

The study’s aims, research questions/hypotheses, and methodology all need to be made much clearer to the reader. It is at times difficult to understand the design decisions taken in planning and executing the study. This needs to be remedied to improve the manuscript.

Validity of the findings

Be careful about what claims you make on the basis of your data analysis. I can’t see all your tables and I’ve not checked your calculations in every case. However, I think your approach to data analysis seems fine. What I would caution against is stating that the research definitely shows there is a causal link. It is safer to state that AO as an intervention may improve outcomes given that there’s likely some variables that cannot be accounted for. Don’t overstate what you can definitely infer from your analysis.

You will be able to find additional details throughout the comments on the returned manuscript itself.

Additional comments

The introduction needs substantially more detail on what the study is intended to accomplish. We need clear aims, research questions/hypotheses, and a rationale. Is this intended to produce recommendations and for whom? Why is this necessary (because the literature suggests that AO could improve learning outcomes)?

The methodology needs to make much clearer why you have selected your research methods and designed the study in this way. Why quantitative analysis? Why an experimental design? What advantages do they confer, why do they answer your RQs better than other methods/designs? We need more detail on everything – who your participants are and why they were selected, why the variables measured are indicators of integrated science process skills, in what way are the participants exposed to advanced organisers? Are there any drawbacks to this design? E.g., could the markers score things differently, would second-marking have been helpful? Are there any ethical issues, what have you done to reduce the influence of bias? I would structure the paper something like this: Research Methods, Data Collection, Sample, Data Analysis, Ethical Considerations. At each stage justify what you have done and why you’ve done it.

More use of literature in the introduction. Though the discussion section includes ample literature, I would just introduce some of it – if even briefly – in a single paragraph in the introduction. You can then state with some justification why there is a research gap and what it is.

The use of the word ‘Crises’ in the title implies that this is a variable that is measured, but it is not. Perhaps just stating ‘during COVID-19’ might be less confusing, or merely omitting that part altogether.

I know there’s a lot here but If I were to recommend prioritising anything, it would be to make your study’s aims clear and to state more overtly why you have designed the research in the way that you have.

---

## Round 0.2 · Major Revisions

Follow review comments and revise.

Reviewer 1 ·

Basic reporting

I am satisfied with the changes made by the author to our findings. The paper can be accepted for publication.

Experimental design

I am satisfied with the changes made by the author to our findings. The paper can be accepted for publication.

Validity of the findings

I am satisfied with the changes made by the author to our findings. The paper can be accepted for publication.

·

Basic reporting

Overall comments:
This revised paper represents a significant improvement on the previous version of it submitted some months ago. The authors' submission of their responses to the feedback has been helpful in identifying these changes and access to their tables and appendices have made evaluation of the research instruments and results more complete.

Taken as a whole, almost every area of this study is vastly improved. The abstract conforms to what is conventionally expected, the introduction is clearer as to aims and objectives, the methods justify to some extent the methods selected, and the conclusion makes more reasonable conclusions and recommendations.

However, there are a few areas where modest-to-significant changes may be made to improve the quality of the paper overall. As things stand, I feel the paper could be published should certain changes be implemented. I’ve outlined some of these on the paper itself and will summarise also below.

Referencing:
The referencing used is extensive and relevant. I’ve highlighted a couple of queries with regards to how you’re citing researcher names in text that may require review. Generally, however, the standard is very high.

Organisation:
The work is logically structured and flows very well throughout. I can see that structural changes have been implemented and this really helps with the flow of the paper.

Experimental design

Suggestions:
1. Extend the review of the literature on AOs in order to make clear the gap in the literature that your research will close. Whilst you indicate that there is sufficient reason to expect AOs to improve science skills, you haven’t given any indication as to why this research is necessary. There are several potential ways of doing this. Critique the literature and state one or some combination of the following: a) not enough research has been carried out with regards to improving science skills themselves; b) not enough research has been carried out in university-level contexts, and/or; c) not enough research has been carried out in KSA. This will tell us why your research is needed.

2. Although the methodological decisions have been expanded, slightly more is needed in this area. I’m thinking especially of the sampling, though potentially also every section if you intend to proceed with using mixed methods, as the qualitative methods have not been described in nearly enough detail. This brings me on to the third area.

3.The qualitative methods I believe require a significant rethink. At the moment, they feel tagged on and there is no real evidence that qualitative analysis has been carried out. I don’t see any attached research plans or evidence of their analysis. Furthermore, the methods themselves are under-described and insufficiently justified. You seem to have used thematic analysis to evaluate the quality of their submitted plans, which isn’t typically how you would use thematic analysis. You either need to rethink the methods of qualitative analysis and go substantially farther in describing, justifying and evidencing it, or dispense with it altogether. I think that perhaps just removing it might be simpler, though if you want to persist with including it, then it needs significant work.

Validity of the findings

The findings section could benefit from slight condensation, while the discussion section could be expanded. Your reporting in the findings section is detailed and clear. However, it occasionally restates the information provided in Table 2 without contributing additional insights. I recommend streamlining this section and focusing more on enhancing the critical discussion in the subsequent section. Identifying a research gap in your literature review and addressing this gap in your discussion will significantly contribute to elevating the quality of your paper

Additional comments

The spelling and grammar are significantly improved upon the last submission and the authors' work in this regard has not gone unnoticed. However, there are still minor tweaks required throughout the paper. I’ve made some minor amendments and some other recommendations. Do not be discouraged by this – this is a piece of academic research and standards for publication are very high, so it is natural for even a native English speaker to receive feedback and guidance on language at this level.

Whilst the above advice might seem extensive, it is important to note that your work has already demonstrated significant improvement. This guidance is simply intended to help you achieve the highest level of quality in your work. There is a lot in the present work that is of a high standard and the changes already made are extensive and ought to be commended.

---

## Round 0.3 · accepted · Accept

I am writing to inform you that your manuscript - The Impact of Advance Organizers in Virtual Classrooms on the Development of Integrated Science Process Skills - has been Accepted for publication. Congratulations!

This is an editorial acceptance; publication is dependent on authors meeting all journal policies and guidelines.

·

Basic reporting

The revised manuscript is now clearly articulated in professional English, enhancing readability and comprehension. I appreciate the authors' efforts to address the language issues previously identified, resulting in a text that flows smoothly and effectively communicates the research undertaken. The background context and literature references are now well-integrated, offering readers a comprehensive understanding of the field. The structure of the article, including figures and tables, is professionally presented, and the data have been made available, ensuring transparency and reproducibility.

Experimental design

The authors have made significant improvements in defining the research question and detailing the experimental design. The revised manuscript now clearly outlines how the research addresses a gap in the current literature, and the methods section has been expanded to provide the necessary detail for replication. The ethical standards and methodological rigor of the study are evident.

Validity of the findings

I am satisfied with the revisions made concerning the validity of the findings. The authors have been careful not to overstate their claims and have presented their analysis in a manner that is statistically sound and controlled. The conclusions are now well-connected to the research question and are supported by the results, demonstrating a clear understanding of the limitations of the study.

Additional comments

The authors have thoroughly addressed my concerns and have made corresponding amendments that significantly improve the manuscript. The clarity of the abstract, the coherence of the conclusions, and the detailed explanation of the research methodology are particularly noteworthy. The authors' diligent response to the review comments is appreciated, and it is clear that careful consideration has been given to each point raised during the review process.

In light of the substantial revisions and the careful attention to detail demonstrated by the authors, I am delighted to suggest to the editorial board that the paper meets the PeerJ criteria and should be accepted.